# Comparative Analysis of Deep Learning Architectures for Macular Hole Segmentation in OCT Images: A Performance Evaluation of U-Net Variants

**DOI:** 10.3390/jimaging11020053

**Published:** 2025-02-11

**Authors:** H. M. S. S. Herath, S. L. P. Yasakethu, Nuwan Madusanka, Myunggi Yi, Byeong-Il Lee

**Affiliations:** 1Department of Industry 4.0 Convergence Bionics Engineering, Pukyoung National University, Busan 48513, Republic of Korea; sewmi96@pukyong.ac.kr; 2Faculty of Technology, Sri Lanka Technological Campus, Padukka 10500, Sri Lanka; lasithy@sltc.ac.lk; 3Digital Healthcare Research Center, Pukyong National University, Busan 48513, Republic of Korea; nuwanv@pknu.ac.kr; 4Division of Smart Healthcare, College of Information Technology and Convergence, Pukyong National University, Busan 48513, Republic of Korea

**Keywords:** optical coherence tomography, segmentation, convolutional neural networks, macular hole

## Abstract

This study presents a comprehensive comparison of U-Net variants with different backbone architectures for Macular Hole (MH) segmentation in optical coherence tomography (OCT) images. We evaluated eleven architectures, including U-Net combined with InceptionNetV4, VGG16, VGG19, ResNet152, DenseNet121, EfficientNet-B7, MobileNetV2, Xception, and Transformer. Models were assessed using the Dice coefficient and HD95 metrics on the OIMHS dataset. While HD95 proved unreliable for small regions like MH, often returning ‘nan’ values, the Dice coefficient provided consistent performance evaluation. InceptionNetV4 + U-Net achieved the highest Dice coefficient (0.9672), demonstrating superior segmentation accuracy. Although considered state-of-the-art, Transformer + U-Net showed poor performance in MH and intraretinal cyst (IRC) segmentation. Analysis of computational resources revealed that MobileNetV2 + U-Net offered the most efficient performance with minimal parameters, while InceptionNetV4 + U-Net balanced accuracy with moderate computational demands. Our findings suggest that CNN-based backbones, particularly InceptionNetV4, are more effective than Transformer architectures for OCT image segmentation, with InceptionNetV4 + U-Net emerging as the most promising model for clinical applications.

## 1. Introduction

Optical Coherence Tomography (OCT) has revolutionized ophthalmology by enabling high-resolution, non-invasive, in-depth analysis and visualization of retinal microstructure.

This imaging modality has become instrumental in diagnosing and managing various retinal pathologies, including age-related macular degeneration, diabetic retinopathy, and glaucoma [1,2]. While OCT’s diagnostic value is well-established, the increasing volume and complexity of image data necessitate automated processing solutions.

Manual retinal layer segmentation in OCT images is time-intensive and subject to observer variability, driving the development of automated segmentation techniques [3]. Although traditional image processing approaches have shown utility, recent advances in machine learning, particularly Convolutional Neural Networks (CNN), have demonstrated superior performance in capturing complex retinal features [4,5]. However, implementing deep learning models for OCT segmentation presents challenges, including computational resource requirements and the need for large, annotated datasets.

Macular hole (MH) detection represents a particular challenge in OCT image segmentation. These retinal defects, characterized by breaks in the macula, can lead to severe visual impairment if left untreated. The accuracy of segmentation algorithms is heavily influenced by image quality, necessitating robust approaches that can perform reliably across various imaging conditions.

This paper utilizes the OIMHS dataset to comparatively assess CNN-based backbones and U-Net on OCT images for automatic retinal segmentation and macular hole detection. We aim to enhance automatic retinal disease diagnosis and mitigate the constraints of OCT image processing in the existing literature. The development of such systems has the prospect of a significant impact on early diagnosis and treatment planning in clinical ophthalmology.

## 2. Background

### 2.1. Related Work

Recent advances in deep learning have significantly transformed OCT image segmentation. This review synthesizes key methodological developments, focusing on publications between 2023 and 2025, identified through systematic searching of “OCT” and “Segmentation” keywords in scholarly databases.

Current research primarily focuses on three main directions: architectural innovations, domain-specific challenges, and data-handling improvements. In architectural innovations, ref. [1] introduced the SASAN system, which enhances segmentation accuracy through improved spectral–spatial information integration, particularly for pathologies like macular edema. Similarly, ref. [5] developed SeqCorr-EUNet, combining U-Net and EfficientNet architectures to achieve superior anterior segment feature segmentation. Ref. [6] utilizes an SFNet, a novel segmentation network that combines a spatial and frequency domain approach and creates the REVIO dataset for retinal OCTA vessel segmentation. Together with REVIO, ROSE, and OCTA-500 datasets, SFNet achieves state-of-the-art results and facilitates the development of accurate diagnostic and therapeutic measures for vascular diseases by enabling their early-stage detection.

Domain-specific challenges have been addressed through various approaches. Refs. [2,3] proposed a multistage approach utilizing EfficientNet, ResNet, and modified Attention U-Net to handle noise and low contrast in retinal OCT images. Notably, ref. [3] introduced a weakly supervised framework using pseudo-volumetric labels generated by a 3D-GDH algorithm, reducing manual annotation requirements while maintaining accuracy. And ref. [7] illustrates a method for semi-supervised segmentation of retinal layers for an image that utilizes gruesomely sparse labeled data in an unlabeled-dominated environment. This newly introduced topology prediction engine lifts the upper boundary on accountable performance. We verify that this method is robust in terms of cross-dataset and cross-acquisition protocol, increasing label efficiency and trustworthiness in managing retinal diseases. Wang et al. [8] present AMSC-Net, a semi-supervised fluid segmentation network in OCT images with a 73.95% Dice score with 5% labeled data. AMSC-Net combines Heterogeneous Architecture Consistency, Multi-label Semantic Consistency Loss, and Anatomy Contour Consistency Loss to enhance segmentation performance. Extensive experiments verify its state-of-the-art performance and clinical usefulness. The work of Ji et al. [9] report Mirrored X-Net, a weakly supervised Geographic Attery (GA) segmentation model in SD-OCT imaging with only image-level labels. We introduce Anisotropic Downsampling for better feature extraction and a new contrastive module for better differentiating between the classes.

Patch-based CNN classifier employs VGG16 and ResNet50 [4] and segments the corneal layers in AS-OCT images by dividing them into smaller patches to be trained further and also to be diagnosed for various corneal diseases. Rough fuzzy discretization has been introduced into deep learning architectures to improve segmentation into components by dealing with noise and variability through a combination of traditional and advanced techniques [10]. Image segmentation remains crucial; where k-means clustering was best emphasized, it could not be complete without edge-based techniques such as the Sobel, Canny, and Robert’s operators; this study also added Wiener filter pre-processing to minimize speckle noise [11]. Additionally, the Canny operator enhanced boundary detection with a multi-point boundary search and convolution kernel changes and yielded more than 98% “perfect” or “good” segmentation in both healthy and AMD-affected individuals [12]. The algorithm also works as a standalone method or an ideal pre-processing step for boundary detection to enhance segmentation reliability.

Domain adaptation and transfer learning for OCT image segmentation from different datasets are essential. A two-stage adversarial learning framework for the unsupervised domain adaptation of segmentation models was introduced by Diao et al. [13] to deal with the above-mentioned domain differences that would enhance segmentation performance. This approach adapts models trained on a certain dataset to another dataset and demonstrates application potential for domain adaptation in medical image analysis.

To overcome the difficulties presented by differences in color, artifacts, and the limited annotated data, ref. [14] proposed a two-step DiffusionDCI framework for the generation and segmentation of Dynamic Cell Imaging (DCI). The first step consists of a Dual Semantic Diffusion Model (DSDM) that would augment the input data, while the second applies diffusion features to a segmentation network with a cross-attention alignment module for increased segmentation accuracy. The results of these experiments show that this method produces high-quality, diverse images that define cancerous regions accurately so that they can be used to improve surgical outcomes. Future work includes improving resolution, refining masks, and using Latent Diffusion Models for better control and stability. Meanwhile, ref. [15] proposes FNeXter, a novel multi-scale feature fusion attention network for fluid segmentation in retinal OCT images. By incorporating ConvNeXt, Transformer, and region-aware spatial attention, FNeXter learns to attend to lesion areas with both global and local features. Its superiority is confirmed by extensive experiments over the state-of-the-art methods for fluid segmentation. The work by [16] introduces the Multi-Scale Aggregation and Location Information Fusion Network (MAPI-Net) as a new model for multi-category segmentation of vulnerable plaque in Intravascular Optical Coherence Tomography (IVOCT) images to solve problems like varying shapes and indistinct boundaries. MAPI-Net enhances segmentation performance through the application of multi-scale features and plaque location distribution. Rigorous experiments confirm its advantage over comparison models and facilitate cardiovascular disease research.

Another strong motivation for improvement in this area has been the release of new datasets, mainly for OCT image segmentation. First introduced by S. He et al. [17], the OIMHS dataset was designed specifically for manual macular hole segmentation tasks in OCT images. The availability of this curated dataset has proved quite instrumental in providing a uniform benchmark for both segmentation model evaluation and training, underlining the importance of excellent annotated data in OCT image processing.

Cao et al. [18] supplemented retinal OCT image segmentation by augmenting their attention mechanism to yield better feature extraction as well as improving model topology-segmentation accuracy through attention layer embedding. Liu et al. [19] have come up with a new method, Feature Pyramid Fusion Network, with a Dynamic Perception Transformer for performing retinal biomarker segmentation that outperforms the conventional methods by exploiting multiscale features. All these improvements show that having attention mechanisms and feature pyramids in a transformer-based model does enhance this segmentation of retinal OCT images. Kumar et al. [20] used this principle and devised a GAN-based process that will synthesize OCT images with vital features for segmentation and keep the information private concerning patients. This shows that GANs can balance privacy and data utility, making them very useful in medical image analysis. Xiao et al. [21] illustrated the use of the EA-Unet model for the segmentation of images of the uterine cavity, which is based on retinal OCT data that deep learning architectures can adapt across different medical imaging processes or applications. In [22], Kugelman et al. enhance OCT retinal and choroidal layer segmentation using GAN-based data augmentation and semi-supervised learning (SSL). Conditional StyleGAN2 generates synthetic patches, and an optimized patch classifier architecture improves performance. This approach demonstrates significant improvement in segmentation accuracy, particularly with sparse data, to the benefit of researchers and clinicians in medical imaging.

Montazerin et al. [23] developed semi-automatic software for the robust segmentation of eight macular layers and the specific detection of retinal pathologies such as diabetic macular edema. The developed method combines Dijkstra’s Shortest Path First algorithm, Live-wire function, and preprocessing operations to improve segmentation accuracy. This approach provides detailed layer segmentation and precise localization of fluid objects with acceptable Dice scores, hence making it more efficient and much less expensive than manual segmentation with high repeatability. Ref. [24] assesses the image segmentations of normal and diseased eyes using proprietary software (Heidelberg Spectralis HRA + OCT, version 1.10.4.0) and cross-platform OCT segmentation software (Orion, version 3.0.0). The results were compared to the ‘golden standard’ of manual segmentation. Cross-platform was considered reliable in research settings for studying the retinal layers as it compares well against manual segmentation and the commonly used proprietary software for both normal and diseased eyes.

Authors of [25] developed a GD-Net, a U-shaped retinal layer and fluid segmentation network in OCT images. Combined with a Fast Fourier Encoder and Multi-scale Graph Convolution module, GD-Net segments accurately and with good across-dataset generalization, which facilitates ophthalmologists to diagnose and monitor Macular Edema (ME).

Authors of [26] propose a general retinal layer segmentation method for OCT images, addressing dataset variation and disease interference. With an improved decoding module, feature channel, position channel attention, and focal loss, it eliminates boundaries, strengthens structural perception, and balances classes. It achieves state-of-the-art performance on five datasets and can aid in ocular disease diagnosis and research.

Cao et al. [27] introduced ScLNet, a deep learning method for accurate segmentation of corneal layers and tear fluid reservoir (TFR) under scleral lenses in OCT images. ScLNet, with 31,360 images as a dataset, achieves high segmentation accuracy and surpasses state-of-the-art methods in the detection of scleral lens, TFR, and cornea boundaries. Deep GD, a novel early glaucoma detection model from retinal fundus images, has been proposed by [28]. It employs preprocessing; CCRS Snap ECNN is used for feature extraction; EfficientNetB4 is used for classification, and Aquila is used for optimization efficiency. It achieves 99.35% classification and segmentation accuracy and is better than existing methods in glaucoma detection.

Ref. [29] proposes an effective method of automatic diabetic retinopathy feature (aneurysm and exudate) segmentation from OCT images. This method, based on a U-Net++ model combined with adaptive thresholding and bagged tree classifiers, achieved superior segmentation performance (87.0% F1-score) and was much better compared to other methods like binary thresholding and watershed. In the work by [30], deep learning methods for retinal OCT image segmentation are reviewed and analyzed to show how they are affected by data variability across different device manufacturers. nnUNet, SAMedOCT, and IAUNet_SPP_CL algorithms are compared over the RETOUCH challenge dataset. nnUNet_RASPP is determined to yield the best fluid segmentation with an 82.3% mean Dice score.

According to the literature review, the reviewed material reflects that significant improvements have been made in deep learning for OCT image segmentation. These contributions have presented a range of approaches, from using powerful neural networks and attention mechanisms to incorporating spectral data along with the spatial. OCT image analysis is continuing to advance due to the creation of new datasets, creative methods, and improved models. These developments promise better diagnostic capabilities and patient outcomes in ophthalmology and other fields. The following Table 1 depicts the insights of the literature.

The studies reviewed have identified a number of methods and major advancements in OCT image segmentation. These vary from methods such as the utilization of spectral and spatial information, the leveraging of weakly supervised frameworks, and the utilization of ensemble learning. All these advancements have progressively improved segmentation accuracy while reducing the need for large-scale manual annotations. Drawing an idea from these findings, our research focuses on the application of these new methods to publicly available ophthalmic image datasets. With the investigation of novel approaches and the creation of more sophisticated segmentation models, we aim to keep pushing the field of ophthalmic imaging forward.

Based on Table 2, which presents Dice coefficient values obtained from various studies using different datasets, we were inspired by these architectures to investigate the segmentation performance of U-Net with its variants. Specifically, we explored variations in its encoder to achieve a statistically significant Dice coefficient.

### 2.2. Selection of Models for Segmentation

By analyzing the existing literature, we have identified that U-Net plays a crucial role in medical image segmentation, particularly in retinal OCT analysis. U-Net’s encoder–decoder structure facilitates effective learning of spatial features, making it a robust choice for segmentation tasks. Moreover, modifications to the underlying architecture can significantly enhance performance by improving feature extraction, addressing class imbalances, and refining boundary delineation.

To further evaluate the effectiveness of U-Net in segmentation tasks, we assessed its performance when combined with various CNN backbones. Specifically, we experimented with DenseNet121, VGG16, VGG19, ResNet152, InceptionNetV4, EfficientNet-B7, MobileNetV2, XceptionNet, Transformer-based U-Net, and the Segresnet, which is an improved architecture of U-Net. These architectures have demonstrated strong capabilities in feature extraction, deep hierarchical representation learning, and generalization across medical imaging datasets. The primary objective of this study is to analyze the segmentation performance of these architectures, particularly in minority-class regions where traditional models often struggle.

The use of a variety of models, such as U-Net and hybrid architectures combined with backbone networks like ResNet, DenseNet, VGG, InceptionNet, EfficientNet, MobileNet, Xception, and Transformer, is well-justified for OCT image segmentation. U-Net is ideal for medical segmentation tasks due to its encoder–decoder structure and skip connections that preserve spatial information. By integrating it with networks like DenseNet121 and ResNet152, the models benefit from deeper feature extraction and residual connections, enhancing the segmentation of complex structures. VGG16/19, with its deep layers, aids in hierarchical feature extraction, while InceptionNetV4’s multi-scale processing captures fine details across diverse pathological cases. EfficientNet-B7 comes with a well-optimized architecture to balance the tree into depth and resolution and, hence, works best in segmentation efficiency, while MobileNetV2 provides a lightweight option for real-time and resource-constrained environments. Also, XceptionNet depth separable convolution can efficiently learn the spatial and channel-wise features, while transformers boost the long-range dependency learning through the attention mechanism and consequently enhance the segmentation accuracy. The combination of different approaches creates room for pre-trained weights that improve feature extraction for noise and low contrast, which, thus, proves adaptable and robust hybrid models for accurate segmentation of OCT images in different clinical and pathological conditions.

## 3. Methodology

### 3.1. Data Acquisition

The OIMHS dataset [17] is a publicly accessible OCT Image Macular Hole Segmentation (OIMHS) dataset that consists of 3859 B-scan images from 119 patients, each with four annotations indicating the retina, macular hole, intraretinal cysts, and choroid. Initially, three junior ophthalmologists performed the annotations that an expert ophthalmologist reviewed to ensure accuracy. This dataset attempts to address a few limitations associated with existing datasets; the available datasets are mainly leaving no publicly accessible database for MH segmentation; the absence of lesion segmentation in most OCT image datasets makes it difficult to draw morphological features of MH manually, while there are only a few datasets on assessing the quality of OCT image, which affects the accuracy of segmentation techniques, indicating the need for evaluating image quality for better performance of segmentation algorithms.

### 3.2. Preprocessing

Segmenting and classifying macular holes in OCT images involve multiple stages, ensuring dependable automated retinal analysis. It starts with obtaining high-resolution retinal OCT images, which offer in-depth structural details to spot conditions such as macular holes from the selected dataset. The dataset comprises high-resolution retinal OCT images, providing detailed structural insights essential for identifying conditions such as macular holes.

The system also resizes the images to a set size (like 256 × 256) using interpolation methods. This creates uniformity for batch processing and makes computations more efficient.

### 3.3. Workflow of This Study

The leading segmentation network, often built on an existing CNN design, handles the first step of identifying macular hole areas. By analyzing segmentation models, the Experts select the best CNN model for breaking down medical images. In this approach, the U-Net model is utilized for segmentation, where the encoder is replaced with a selected CNN to perform feature extraction, as shown in Figure 1a. The decoder architecture remains unchanged. This modification aims to enhance the model’s ability to extract relevant features while maintaining the original U-Net structure for effective segmentation, leveraging the strengths of the chosen CNN for feature extraction. On the other hand, Segresnet also uses a segmentation model to observe performance variation.

To train this model, we need to optimize the segmentation network with our training dataset, which we split into training and validation sets. Segmentation performance is measured by some sort of loss function, such as Categorical Cross-Entropy or Dice Loss. An optimizer such as Adam helps carry out the gradient descent to minimize this loss. We would train this model in such a way that with several epochs, weights are modified so that the loss decreases. Once trained, the model is used to predict segmentation masks on new retinal images, thereby highlighting the location of macular holes. The following Figure 1b shows the workflow of this study.

### 3.4. Training Process

Effective training of the segmentation model involves the definition of an appropriate loss function and the selection of suitable optimization algorithms. To do this, one uses Categorical Cross-Entropy Loss as the measure of the discrepancy between predicted segmentation masks and actual labels, quantifying the performance of this model. Minimizing this loss function with the Adam optimization algorithm makes the adjustments necessary in the parameters of this model for improved accuracy. Training is performed with a batch size of 20, in 20 epochs, to balance computational efficiency with allowing for enough learning time for the model to really learn from the data and generalize well on new images. The 0.8 split ratio extracts the training dataset from the total images.

### 3.5. Evaluation Metrics

Dice Coefficient (Dice Similarity Coefficient or DSC)

The Dice Coefficient is a measure of overlap similar to the Intersection over Union (IoU); however, it places greater emphasis on the amount of correct prediction: two times the overlapping area in the total number of pixels in both predicted and ground truth regions. By considering both precision and recall, the Dice score gives a measure of image segmentation so that higher scores are better and lie in the range of 0 to 1, with a maximum score of 1 denoting perfect prediction. It can be especially beneficial for medical image analysis because it can tolerate imbalances in the classes. And because it works well for small differences between predicted masks, it is ideal for applications with small differences in mask predictions.(1)$$Dice\;Coefficient=\frac{2\times True\;Positive}{2\times True\;Positive+False\;Positive+False\;Negative}$$

Harsdorf Distance at 95th percentile (HD95 Score)

This metric evaluates the similarity between two contours or segmentation, particularly in medical imaging. It calculates the Harsdorf Distance but focuses on the 95th percentile, making it more robust to outliers than the maximum Harsdorf Distance.

Let A and B be two sets of points representing the surfaces of two objects (ground truth and predicted segmentation):

Forward Distance: For each point aϵA, compute the minimum distance to any point bϵB:(2)$$d\left(a,B\right)=\underset{\mathrm{b\epsilon B}}{\min}\left\|a-b\right\|$$Reverse Distance: For each point bϵB, compute the minimum distance to any point aϵA:(3)$$d\left(b,A\right)=\underset{\mathrm{a\epsilon A}}{\min}\left\|b-a\right\|$$Combine Distances: Collect all minimum distances:(4)$$D=\left\{d\left(a,B\right):\left.a\epsilon A\right\}\cup \left\{d\left(b,A\right):\left.b\epsilon B\right\}\right.\right.$$Compute 95th Percentile: Find the 95th percentile of D:(5)$$HD95\left(A,B\right)={percentile}_{95}(D)$$

## 4. Experiments and Results

### 4.1. Comparative Analysis of Evaluation Metrices Across the Classes

The results highlight clear trends in the segmentation performance of different models across the four classes: MH; Retina; Choroid; and Intraretinal cysts (IRC), as portrayed in Figure 2. It is observed that the models, such as InceptionNetV4 + U-Net, VGG16 + U-Net, and VGG19 + U-Net, consistently yield a high Dice Coefficient for all layers but with more fluctuation for other layers and layers with lower and medium difficulty segmented with scores greater than 0.98 by the model. The MH class presents moderate fluctuation, where all the models present good results, while IRC is the most demanding class, presenting high declines in Dice Coefficients, especially the Transformer + U-Net, which has a poor performance in perceiving the details of the IRC graphs. From this, it may be seen that the CNN-based U-Net variants capable of selecting stronger backbones, like InceptionNet and VGG, are more advantageous for general segmentation tasks, and the Transformer + U-Net model needs further delicate tuning or needs to go hybrid with CNNs. In particular, presented results indicate that the average performance for the IRC channel is systematically lower than for the rest of the classes, which suggests that further development of the feature extraction techniques, specific preprocessing, and probably more resistant training algorithms are needed to properly handle this particular class.

Table 3 HD95 scores from the different models that segment MH, retina, choroid, and IRC regions in OCT images. Results show that for the category of MH and IRC, all values come out as ‘nan’. This is mostly due to the small sizes of these regions, which inhibit proper segmentation by the models. For retina segmentation, the best performer according to the HD95 score is InceptionNetV4 + U-Net, with the lowest score, whereas EfficientNet-b7 + U-Net has the highest, indicating relatively worse performance. The same trend of results is shown for choroid segmentation, where InceptionNetV4 + U-Net is the best, and MobilenetV2 + U-Net scored the least. Transformer + U-Net shows a very high HD95 score (retina and choroid), which indicates that it performs poorly among other models, notwithstanding its promising role in segmentation tasks enhanced with attention mechanisms. VGG-inspired models, VGG16 + U-Net and VGG19 + U-Net, achieved average performance in the range of 1.05–1.9 for retina and choroid. Despite being considered advanced architectures, EfficientNet-b7 + U-Net and MobilenetV2 + U-Net went ahead to produce higher scores, especially in the choroid. On the performance highlights, InceptionNetV4 + U-Net is the best model for both retina and choroid segmentation. Recommendations include the suggestion that InceptionNetV4 + U-Net be used since it has been consistent with doing well in these tasks. Further research on the high HD95 scores of Transformer + U-Net would be worthwhile, as it could help optimize its performance.

### 4.2. Comparative Analysis of Computational Resources

The results of different models concerning B-scan image segmentation are compared in Table 4. From the model analysis, it is deduced that InceptionNetV4 + U-Net, with the lowest validation loss, has the highest Dice coefficient and has an HD95 score that is reasonably comparable, ensuring excellent generalization, segmentation accuracy, and boundary prediction. MobileNetV2 + U-Net and VGG16 + U-Net had similarly high segmentation performance with dice coefficients, respectively, though MobileNetV2 + U-Net had low boundary accuracy, as reflected by a very high HD95 score. Models, such as ResNet152 + U-Net and VGG16 + U-Net, perform very well in terms of segmentation accuracy with Dice scores, respectively, though their higher respective validation losses are probably related to overfitting or poor generalization. U-Net and Segresnet offer balanced performances, with solid validation losses, good Dice scores, and reasonable HD95 values, making them reliable options for segmentation tasks. Overall, InceptionNetV4 + U-Net provides the best balance of accuracy and generalization, while VGG16 + U-Net and ResNet152 + U-Net could benefit from further optimization to improve generalization. Overall, InceptionnetV4 + U-Net is the most effective model, combining low validation loss and high segmentation accuracy. Figure 3 illustrates the above results across the models.

The performance metrics in Table 5 demonstrate the distinct trade-off between the U-Net variants. Standard U-Net presents low GFLOPS with a minimum GPU memory consumption and is the fastest, with an inference time; hence, this architecture can be considered in resource-constrained environments. The Segresnet demonstrated efficiency but slightly higher GFLOPS and memory consumption. A lightweight model like MobileNetV2 + U-Net reaches a balance, having a low parameter count and moderate memory usage while, at the same time, ensuring fast inference. On the other extreme, models heavy with parameters, like ResNet152 + U-Net and EfficientNet-B7 + U-Net, have large feature extraction capabilities but require high GPU memory and are somewhat slower with 0.0262 and 0.0514 s inferences. Transformer + U-Net, with state-of-the-art transformer architecture, shows higher latency when the computation complexity is relatively high GFLOPS. The VGG16 + U-Net and VGG19 + U-Net models balance efficiency with computational cost since they have high GFLOPS while keeping their inference speed fast. DenseNet121 + U-Net and InceptionNetV4 + U-Net are balanced with moderate GFLOPS, number of parameters, and inference times. For overall performance, U-Net and Segresnet are good for lightweight applications, while ResNet152 + U-Net, EfficientNet-B7 + U-Net, and Transformer + U-Net are to be cast in high-performance, depending on the resource constraint trade-offs desired by a particular application. Figure 4 portrays the results of the performance comparison. The bubble size indicates the number of parameters.

## 5. Discussion

This study explored several segmentation models of OCT imaging across different anatomical regions, highlighting the challenges and trade-offs arising in clinical applications. CNN-based models, especially the variants of U-Net, were very strong in segmenting larger regions such as the retina and choroid. Examples include InceptionNetV4 + U-Net, VGG16 + U-Net, and VGG19 + U-Net, which were all high-performing in Dice Coefficients such as 0.9653, 0.9679, and 0.9659, respectively, while U-net obtained 0.9593. These have almost always posed a challenge for small, intricate regions, which include MH and IRC. This underlines an intrinsic limitation in capturing finer details by CNNs because of their resolution sensitivity.

Interestingly, InceptionNetV4 + U-Net had impressive performances for intricate textures and boundaries; the identification of MH region evaluated in Dice Coefficients as 0.9494, hence proving to be one of the best candidates for high precision tasks such as surgical interventions. Its performance is well-balanced, and it could serve as a reliable model in a clinical environment.

On the one hand, even with the utilization of advanced attentions for its global feature extraction, Transformer + U-Net achieved poor performance in segmentation tasks (0.78245 in Dice Coefficients and 9.7801 in HD95 percentile) with excessive increases in computational expenses, hereby this result underpins the limitation from the Transformers itself for OCT Segmentation without explicit techniques, such as integration with local feature representations or domain-specific pretrained models.

Efficiency is the key issue for clinical applications with real-time segmentation. Some of the models, like U-Net and Segresnet, were quite computationally efficient, balancing low memory with low processing power (261.51 MB, 276.61 MB in GPU memory usage) and acceptable accuracy (0.9593, 0.8830 in Dice Coefficients and 1.9987, 1.9274 in HD95 percentile). Among all, the MobileNetV2 + U-Net had the lowest computational requirement and was, thus, very suitable for real-time applications on resource-constrained devices. In contrast, ResNet152 + U-Net and EfficientNet-B7 + U-Net had high computational costs with very good feature extraction. It further brings out that preprocessing and multi-scale approaches can improve segmentation, especially in small structures. The hybrid architecture combining local feature sensitivity from CNNs with global attention in Transformers may lead to filling the performance gap.

## 6. Conclusions

This study provides significant insights into the trade-offs between segmentation accuracy and computational efficiency for OCT imaging models. Within CNN-based architectures, InceptionNetV4 + U-Net proved to be a reliable model for the segmentation of larger anatomical regions like the retina and choroid, offering a balance of precision and dependability for clinical applications. However, the limitation of this, along with other CNN models to segment smaller and complex regions of MH and IRC, draws the interest of researchers toward model improvement. Improved preprocessing and advanced feature extraction, along with multi-scale methods, are highly essential to solving these challenges.

It turns out that U-Net and MobileNetV2 + U-Net are the best solutions for real-time applications, especially MobileNetV2 + U-Net, since it is an effective model in a resource-constrained setting with fewer parameters and minimal computational cost.

While the combination of Transformer with U-Net is a very promising direction in terms of the integration of global attention mechanisms, its present limitations in accuracy, computational efficiency, and performance generally indicate extensive refinement in this direction. Hybrid architectures that could combine the strength of CNN with that of the Transformer, aided by domain-specific pretraining along with adaptive strategies for attention, may unlock much better performance in future implementations. Eventually, it shows that such choices of models bear clinical consideration–requirement trade-offs around accuracy in segmentation, efficiency of computation, and constraint consideration against particular tasks. As of yet, the Dice coefficient is an optimum choice for the assessment of fine structure segmentations, as it has handled a number of fine regions by providing the detail on overlaps in OCT image-processing applications.

## Figures and Tables

**Figure 1 jimaging-11-00053-f001:**
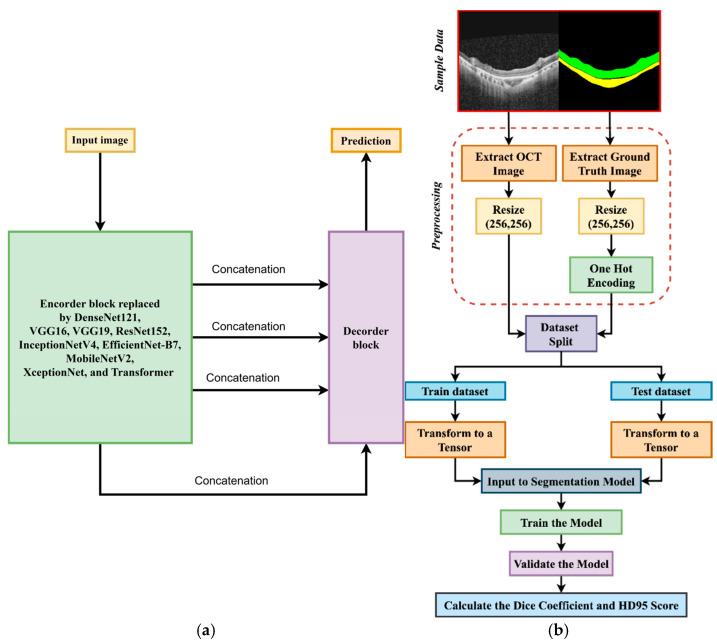
(**a**) U-net replaced by a CNN encoder; (**b**) Workflow of this study.

**Figure 2 jimaging-11-00053-f002:**
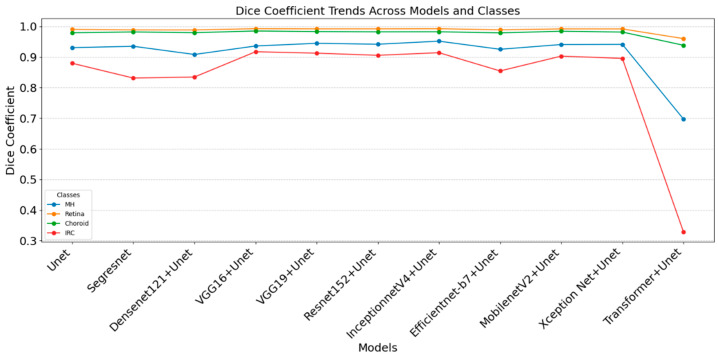
Dice coefficient trends.

**Figure 3 jimaging-11-00053-f003:**
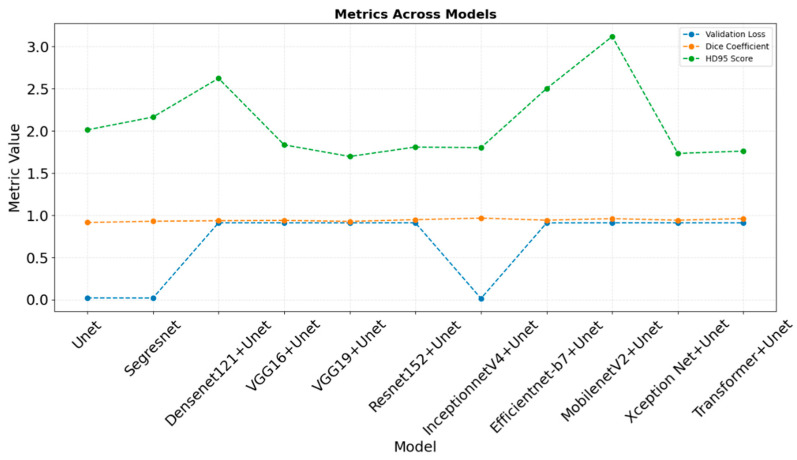
Metrics across models.

**Figure 4 jimaging-11-00053-f004:**
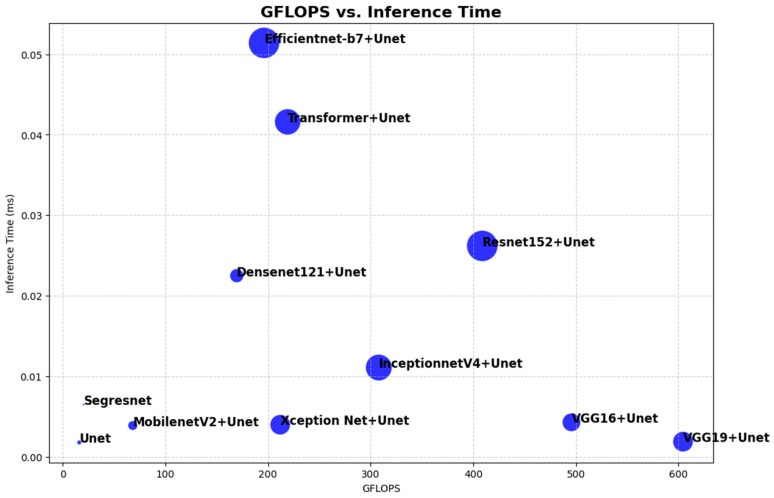
Performance comparison.

**Table 1 jimaging-11-00053-t001:** Insights of the literature on OCT segmentation.

Work	Methodology	Findings
Huang et al. [1] SASAN	Combines spectral and spatial features for retinal OCT segmentation.	Improved retinal layer delineation and segmentation reliability.
Sampath Kumar et al. [2] Multistage Approach	EfficientNet, ResNet, and Attention U-Net for noisy, low-contrast OCT images.	Significant accuracy improvement; reduced noise impact.
Niu et al. [3] 3D-GDH algorithm	Weakly supervised framework with 3D-GDH algorithm for pseudo-labeling.	Reduced annotation needs while maintaining accuracy.
Niu et al. [4] Patch-based CNN Classifier	Trains CNNs (VGG16, ResNet50) on patches for corneal layer segmentation.	Enhanced segmentation accuracy; aids in corneal disease diagnosis.
Fang et al. [5] SeqCorr-EUNet	Combines U-Net and EfficientNet for anterior segment OCT segmentation.	Outperformed prior methods with superior accuracy.
Li et al. [6] SFNet	Combines spatial-frequency domains.	Achieved state-of-the-art accuracy, improves early vascular disease detection.
Fazekas et al. [7] SD-LayerNet	Semi-supervised segmentation leverages sparse labels, topology prediction.	Boosts cross-dataset robustness, label efficiency, and retinal disease management.
Wang et al. [8] AMSC-Net	Semi-supervised fluid segmentation, uses consistency losses for enhancement.	Achieves 73.95% Dice with 5% labeled data, clinical usefulness.
Ji et al. [9] Mirrored X-Net	Uses weak supervision, anisotropic downsampling, contrastive module.	Improved feature extraction, better class differentiation in GA segmentation.
Chen et al. [10] Rough Fuzzy Discretization	Adds rough fuzzy logic to deep learning for noisy OCT segmentation.	Increased segmentation robustness and accuracy.
Sheeba et al. [11] K-means Clustering	Applies clustering techniques with Wiener filter preprocessing.	High accuracy; minimized MSE and maximized PSNR.
Liu et al. [12] Canny operator	An improved algorithm that adds a multi-point boundary search step based on the original method and adjusts the convolution kernel.	Beneficial for use alone or in combination with other methods in initial boundary detection.
Diao et al. [13] Two-stage Adversarial Learning	Domain adaptation via adversarial learning for OCT datasets.	Enhanced cross-dataset segmentation performance.
Yang et al. [14] DiffusionDCI	Dual Semantic Diffusion Model for generating and segmenting DCI with cross-attention.	Accurate segmentation and high-fidelity image generation.
Niu et al. [15] FNeXter	Combines ConvNeXt, Transformer, spatial attention for fluid segmentation.	Outperforms state-of-the-art methods in retinal OCT fluid segmentation.
Su et al. [16] MAPI-Net	Uses multi-scale features and location fusion for plaque segmentation.	Outperforms comparison models, aids cardiovascular disease research and diagnosis.
S. He et al. [17] OIMHS Dataset	Released dataset for macular hole segmentation in OCT images.	Provided a benchmark for training and evaluation.
Cao et al. [18] Transformer-based Attention	Transformer-based attention for retinal OCT segmentation.	Improved feature extraction and accuracy.
Liu et al. [19] Feature Pyramid Fusion Network	Multiscale features with Dynamic Perception Transformer for biomarker segmentation.	Robust accuracy; outperformed traditional methods.
Liu et al. [20] GAN-based Privacy Approach	Uses GANs to synthesize privacy-preserving OCT images.	Balanced privacy and segmentation performance.
Xiao et al. [21] EA-UNet Adaptation	Adapted EA-UNet for uterine cavity segmentation from retinal OCT data.	Demonstrated model flexibility for diverse imaging tasks.
Kugelman et al. [22] Conditional StyleGAN2	GAN-based augmentation, semi-supervised learning for enhanced segmentation.	Improves accuracy with sparse data, aids medical imaging research.
Montazerin et al. [23] Livelayer	Dijkstra’s Shortest Path First (SPF) algorithm and the Livewire function together with some preprocessing operations on the segmented images.	Detailed layer segmentation and fluid localization with reduced manual effort.
Alex et al. [24] Comparing automated retinal layer segmentation	Retinal segmentation compared proprietary and cross-platform software against manual grading and layer volumes.	Cross-platform software excels in specific layer segmentation and correlates well with manual standards.
Cao et al. [25] GD-Net	Uses FFT encoder, graph convolution for segmentation.	Achieves accuracy, cross-dataset generalization, aids Macular Edema diagnosis.
Hao et al. [26] General segmentation method	Improved decoder, attention mechanisms, focal loss for segmentation.	Excels on five datasets, aids ocular disease diagnosis, research.
Cao et al. [27] ScLNet	Deep learning for scleral lens, corneal, TFR segmentation.	Surpasses state-of-the-art methods, achieves high segmentation accuracy.
Geetha et al. [28] DEEP GD	CCRS ECNN, EfficientNetB4, Aquila optimization	Achieves 99.35% accuracy, outperforms existing glaucoma detection methods.
Tanthanathewin et al. [29] Method based on U-Net++	U-Net++ with adaptive thresholding, bagged tree classifiers for segmentation.	Achieved 87.0% F1-score, outperforms binary thresholding, watershed methods.
Ndipenoch et al. [30] Algorithm comparison	Reviewed nnUNet, SAMedOCT, IAUNet_SPP_CL on RETOUCH dataset.	nnUNet_RASPP achieves best fluid segmentation.

**Table 2 jimaging-11-00053-t002:** Dice Coefficient Values Reported in the Literature for Segmentation Across Various Datasets.

Work	Dice Coefficient	Dataset
Huang et al. [1]	93.53 ± 17.35	OIMHS DATASET
Sampath Kumar et al. [2]	82.25 ± 0.74%	Duke SD-OCT
Li et al. [6]	83.85	Retinal Vessels Images in OCTA (REVIO) dataset
	76.7	ROSE: A Retinal OCT Angiography Vessel Segmentation Dataset and New Model
	82.85	OCTA—500
Wang et al. [8]	73.95%	Private dataset
Chen et al. [10]	0.97	Private dataset
Yang et al. [14]	0.8302	DCI dataset
Niu et al. [15]	82.33 ± 0.46	RETOUCH
Cao et al. [18]	0.845	DUKE DME
Liu et al. [19]	80.23	Local biomarker dataset
Cao et al. [25]	0.839	DUKE DME
	0.826	Peripapillary OCT
	0.872	RETOUCH
Hao et al. [26]	82.64	MGU dataset
	87.42	DUKE
	91.95	NR206
	94.32	OCTA500
	89.55	Private dataset
Cao et al. [27]	96.50%	Private dataset
Ndipenoch et al. [30]	82.3%	RETOUCH

**Table 3 jimaging-11-00053-t003:** HD95 score across the models and classes.

Model	MH	Retina	Choroid	IRC
U-Net	nan	1.32069862	1.95789254	nan
Segresnet	nan	1.5654013	1.5654013	nan
Densenet121 + U-Net	nan	1.96263492	2.11148545	nan
VGG16 + U-Net	nan	1.05933894	1.7365639	nan
VGG19 + U-Net	nan	1.06198439	1.90753952	nan
Resnet152 + U-Net	nan	1.09918057	1.77921848	nan
InceptionnetV4 + U-Net	nan	1.0417409	1.81851334	nan
Efficientnet-b7 + U-Net	nan	2.1089202	3.18990847	nan
MobilenetV2 + U-Net	nan	1.88089567	3.71487823	nan
Xception Net + U-Net	nan	1.11548322	1.89977866	nan
Transformer + U-Net	nan	15.29405853	6.77490137	nan

**Table 4 jimaging-11-00053-t004:** Predictions across the models.

Original	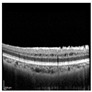	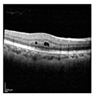	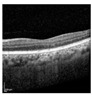	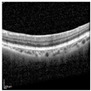	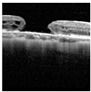
Ground truth	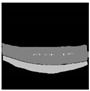	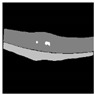	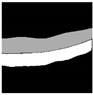	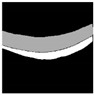	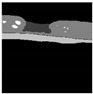
U-Net	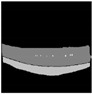	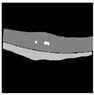	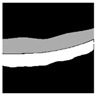	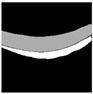	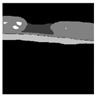
Segresnet	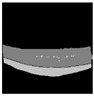	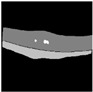	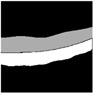	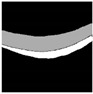	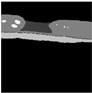
Densenet121 + U-Net	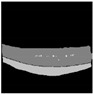	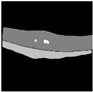	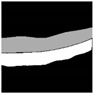	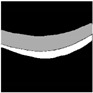	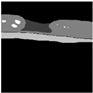
VGG16 + U-Net	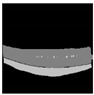	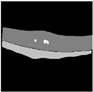	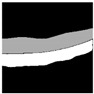	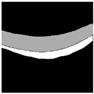	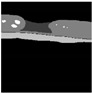
VGG19 + U-Net	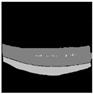	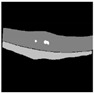	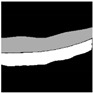	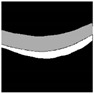	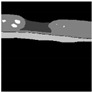
Resnet152 + U-Net	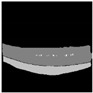	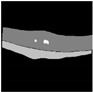	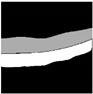	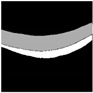	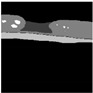
InceptionnetV4 + U-Net	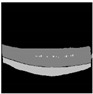	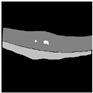	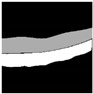	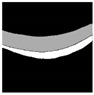	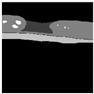
Efficientnet-b7 + U-Net	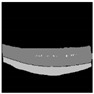	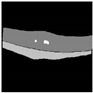	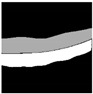	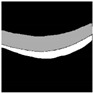	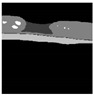
MobilenetV2 + U-Net	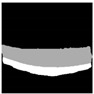	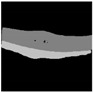	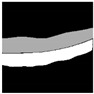	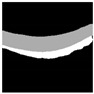	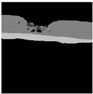
Xception Net + U-Net	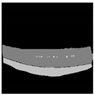	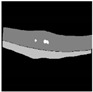	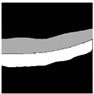	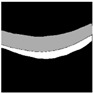	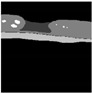
Transformer + U-Net	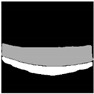	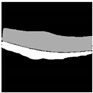	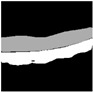	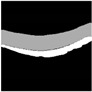	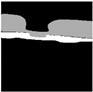

**Table 5 jimaging-11-00053-t005:** Performance Comparison of Segmentation Architectures.

Model	GFLOPS	Number of Parameters	GPU Memory Usage	Inference Time per Batch (Batch Size = 20)
U-Net	16.13	1,626,796	261.51 MB	0.0018 s
Segresnet	20.34	395,157	276.61 MB	0.0065 s
Densenet121 + U-Net	169.63	13,608,213	1021.69 MB	0.0225 s
VGG16 + U-Net	495.70	23,748,821	1568.32 MB	0.0043 s
VGG19 + U-Net	604.41	29,058,517	1644.84 MB	0.0019 s
Resnet152 + U-Net	408.78	67,157,461	1857.09 MB	0.0262 s
InceptionnetV4 + U-Net	307.98	48,792,501	1544.98 MB	0.0111 s
Efficientnet-b7 + U-Net	196.17	67,095,909	1942.01 MB	0.0514 s
MobilenetV2 + U-Net	68.25	6,629,525	726.39 MB	0.0039 s
Xception Net + U-Net	211.92	28,769,981	1195.59 MB	0.0040 s
Transformer + U-Net	219.18	47,353,429	1391.86 MB	0.0416 s

## Data Availability

The publicly shared datasets are available at https://springernature.figshare.com (accessed on 3 May 2024). The codes are available from the author (H.M.S.S.H.; sewmiherath@gmail.com) upon request.

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
