# Peer review of "Comparative Analysis of Deep Learning Architectures for Macular Hole Segmentation in OCT Images: A Performance Evaluation of U-Net Variants"

_2313-433X, 2025, doi:10.3390/jimaging11020053_

Round 1

Reviewer 1 Report

Comments and Suggestions for Authors

This paper presents a comparative analysis of deep learning architectures on OCT images for Macular Hole Segmentation and evaluation of performances of U-Nets. It is a nice paper. The authors successfully presented their research and well written. There will be some updates that are needed to improve the quality of the paper. The authors used 17 reference papers in total. The number of references should be at least more than 30. I would like the authors to add more references and give their details. When the workflow is checked, the dice coefficient and Harsdorf Distance 95th percentile score should be given and discussed at the end of the paper in discussions and conclusions.  The authors should discuss the numerical results in these sections as well. Some minor improvements are needed.

Author Response

Comments 1: There will be some updates that are needed to improve the quality of the paper. The authors used 17 reference papers in total. The number of references should be at least more than 30. I would like the authors to add more references and give their details.

Response : We appreciate the reviewer's observation regarding the limited number of references. We have substantially expanded our literature review by incorporating additional relevant references, particularly focusing on recent developments in medical image segmentation and deep learning architectures for OCT image analysis. The expanded reference list now provides a more comprehensive overview of the current state-of-the-art in this field. The related work section has been enhanced to include these new references, strengthening the theoretical foundation of our research.

Comments 2: When the workflow is checked, the dice coefficient and Harsdorf Distance 95th percentile score should be given and discussed at the end of the paper in discussions and conclusions. The authors should discuss the numerical results in these sections as well

Response : We thank the reviewer for this valuable suggestion. We have thoroughly revised the discussion and conclusion sections to include a detailed analysis of the quantitative results. Specifically, we have added comprehensive discussions of the Dice coefficient and Hausdorff Distance (95th percentile) scores across different architectures. The numerical results have been contextualized within the broader framework of our research objectives, and we have included comparative analyses with existing literature. These additions provide deeper insights into the performance and effectiveness of our proposed approach.

Reviewer 2 Report

Comments and Suggestions for Authors

Line 16: Molecular Hole (MH) abbreviations should have capital letters. Like Molecular Hole (MH).

Line 18: There is an error in VGG/19. It should be written as VGG19.

Line 46: A space should be left before [4,5].

Line 67: A space should be left before [1].

Line 71: A space should be left before [2,3].

Line 91: A space should be left before [9].

Line 110: A space should be left before [12].

Line 112: A space should be left before [13].

Line 130: A space should be left after [17].

Line 145: Edit the table. Justify left and right.

Line 145: Leave a space after [8].

Line 145: Leave a space after [17].

Line 174: Give reference to the dataset.

Line 229: Center the equation.

Line 249: You have given a graph directly after the title. You need to write an introduction.

Line 297: In this table, you can mark the differences to make the differences between the models more obvious. Show in a different table.

Author Response

Comments 1: 

  • Line 16: Molecular Hole (MH) abbreviations should have capital letters. Like Molecular Hole (MH).
  • Line 18: There is an error in VGG/19. It should be written as VGG19.
  • Line 46: A space should be left before [4,5].
  • Line 67: A space should be left before [1].
  • Line 71: A space should be left before [2,3].
  • Line 91: A space should be left before [9].
  • Line 110: A space should be left before [12].
  • Line 112: A space should be left before [13].
  • Line 130: A space should be left after [17].
  • Line 145: Edit the table. Justify left and right.
  • Line 145: Leave a space after [8].
  • Line 145: Leave a space after [17].
  • Line 174: Give reference to the dataset.
  • Line 229: Center the equation.
  • Line 249: You have given a graph directly after the title. You need to write an introduction.

Response : We thank the reviewer for their meticulous attention to detail. We have implemented all suggested formatting corrections as follows:

  • Standardized the abbreviation format for Macular Hole (MH) with proper capitalization
  • Corrected the model notation from VGG/19 to VGG19
  • Added appropriate spacing before all citations [1-5, 9, 12, 13]
  • Added spacing after citations [17, 8]
  • Applied left and right justification to Table 1
  • Added the dataset reference in Line 174
  • Centered the equation in Line 229
  • Added introductory text before the graph following Line 249
  • All citation formatting has been standardized throughout the manuscript for consistency

Comments 2: Line 297: In this table, you can mark the differences to make the differences between the models more obvious. Show in a different table.

Response : We appreciate the suggestion to enhance the clarity of model comparisons. To address this, we have reorganized the presentation of results as follows:

  • Figure 2 now presents the class-wise predictions for each model
  • Figure 3 illustrates the overall Dice coefficient comparisons
  • Table 3 has been redesigned to highlight the performance differences between models, with key metrics emphasized for easier comparison
  • We have added visual emphasis (e.g., bold text, highlighting) to highlight significant differences between model performances

These changes improve the readability and make the comparative analysis more accessible to readers.

Reviewer 3 Report

Comments and Suggestions for Authors

This paper gives a comparative Analysis of Deep Learning Architectures for Macular Hole Segmentation in OCT Images.
The studies are comprehensive and the results are useful for the field of deep learning architecture for macular hole segmentation in OCT images.

Author Response

Comments 1: This paper gives a comparative Analysis of Deep Learning Architectures for Macular Hole Segmentation in OCT Images. The studies are comprehensive and the results are useful for the field of deep learning architecture for macular hole segmentation in OCT images.

Response: We sincerely appreciate the reviewer's positive assessment of our comparative analysis. We aimed to provide a comprehensive evaluation of deep learning architectures for Macular Hole segmentation in OCT images. The confirmation that our results contribute meaningfully to the field is encouraging. We are grateful for the reviewer's recognition of the study's comprehensiveness and practical utility.

Reviewer 4 Report

Comments and Suggestions for Authors

The manuscript compares U-Net variants with different backbone architectures for macular hole (MH) segmentation in optical coherence tomography (OCT) images.

However, some points can be improved:

- The abstract and the introduction discuss different approaches. If, in the abstract, the purpose of the paper was the comparison of various architectures, a novel DL network is proposed in the introduction. Please clarify and improve both sections accordingly.

- You have to clarify your research's aim at line 151. Your task is to perfect the models you described before (all or some of them) ? What kind of new dataset will your research produce?

- As you wrote, it is interesting to see an original image and a high-quality OCT image that the preprocessing step produces.

- As the workflow diagram describes, you should also introduce the test data set in the text. What % of the data have you used for this task?

- The background work should discuss previous works regarding the architecture you selected in more depth.

- A result comparison with previous works is needed in terms of the same evaluation metrics 

- How did you optimize the existing approaches? More discussion of this in the workflow section

Best regards,

Author Response

Comments 1: The abstract and the introduction discuss different approaches. If, in the abstract, the purpose of the paper was the comparison of various architectures, a novel DL network is proposed in the introduction. Please clarify and improve both sections accordingly.

ResponseWe appreciate the reviewer's observation regarding the inconsistency between the abstract and introduction. We have revised both sections to clearly articulate that our primary objective is to conduct a comparative analysis of various deep learning architectures for Macular Hole segmentation. The introduction (Line 54) now explicitly states this purpose and aligns with the abstract.

Comments 2: You have to clarify your research's aim at line 151. Your task is to perfect the models you described before (all or some of them) ? What kind of new dataset will your research produce?

Response: Thank you for highlighting the need for clarity regarding our research aims. We have expanded Line 212 to explicitly state that our primary objective is to evaluate and compare the segmentation performance of different U-Net variants in Macular Hole identification. We have also detailed how each architecture was selected based on its theoretical advantages for medical image segmentation.

Comments 3: As you wrote, it is interesting to see an original image and a high-quality OCT image that the preprocessing step produces.

Response: We appreciate the interest in image preprocessing. As clarified in Line 238, our dataset consists of high-quality OCT images that did not require extensive preprocessing. We have added example images to illustrate the inherent quality of our dataset.

Comments 4: As the workflow diagram describes, you should also introduce the test data set in the text. What % of the data have you used for this task?

Response: We have clarified the data distribution in Line 325, specifying that 80% of the total images were allocated to the training set, with the remaining 20% split between validation and testing. This follows standard practices in deep learning research.

Comments 5: The background work should discuss previous works regarding the architecture you selected in more depth.

Response: We have substantially expanded the background section to provide a more comprehensive review of each selected architecture. The justification for each chosen architecture is now supported by relevant literature and previous successful applications in medical image segmentation.

Comments 6: A result comparison with previous works is needed in terms of the same evaluation metrics

Response: As suggested, we have added a detailed comparison with previous works in Line 221, specifically focusing on Dice coefficient metrics. This comparison provides context for our results within the existing literature.

Comments 7: How did you optimize the existing approaches? More discussion of this in the workflow section

Response: We have enhanced the workflow section (Line 287) to detail our architectural modifications, specifically explaining how we modified the U-Net backbone and our rationale for these changes. The optimization process and its impact on performance are now thoroughly documented.

Round 2

Reviewer 4 Report

Comments and Suggestions for Authors

I insist on making more citations in the Related Work section. In the rest, the manuscript looks fine.

Best regards,